# Shared Care and Virtual Clinics for Glaucoma in a Hospital Setting

**DOI:** 10.3390/jcm10204785

**Published:** 2021-10-19

**Authors:** Anne-Sophie Simons, Julie Vercauteren, João Barbosa-Breda, Ingeborg Stalmans

**Affiliations:** 1Department of Ophthalmology, University Hospitals Leuven, 3000 Leuven, Belgium; ingeborg.stalmans@uzleuven.be; 2Faculty of Medicine, KU Leuven, 3000 Leuven, Belgium; julie.vercauteren@student.kuleuven.be; 3Research Group Ophthalmology, Department of Neurosciences, KU Leuven, 3000 Leuven, Belgium; joao_breda@hotmail.com; 4Cardiovascular R&D Center, Faculty of Medicine of the University of Porto, 4200-319 Porto, Portugal; 5Department of Ophthalmology, Centro Hospitalar e Universitário São João, 4200-319 Porto, Portugal

**Keywords:** glaucoma, ocular hypertension, outpatient clinic, shared care, allied health personnel, collaborative care, patient care, virtual clinic, virtual system, user computer interface

## Abstract

Glaucoma patients require lifelong management, and the prevalence of glaucoma is expected to increase, resulting in capacity problems in many hospital eye departments. New models of care delivery are needed to offer requisite capacity. This review evaluates two alternative schemes for glaucoma care within a hospital, i.e., shared care (SC) and virtual clinics (VCs), whereby non-medical staff are entrusted with more responsibilities, and compares these schemes with the “traditional” ophthalmologist-led outpatient service (standard care). A literature search was conducted in three large bibliographic databases (PubMed, Embase, and Trip), and the abstracts from the prior five annual meetings of the Association for Research in Vision and Ophthalmology were consulted. Twenty-nine were included in the review (14 on SC and 15 on VCs). Patients with low risk of vision loss were considered suitable for these approaches. Among the non-medical staff, optometrists were the most frequently involved. The quality of both schemes was good and improved with the non-medical staff being trained in glaucoma care. No evidence was found on patients feeling disadvantaged by the lack of a doctor visit. Both schemes increased the hospital’s efficiency. Both SC and VCs are promising approaches to tackle the upcoming capacity problems of hospital-based glaucoma care.

## 1. Introduction

Glaucoma is the leading cause for irreversible visual loss worldwide [1]. Periodic assessments are necessary to detect progression at an early stage and to adjust treatment in order to prevent further damage. Once diagnosed with glaucoma, even when asymptomatic, the patient requires lifelong management [2].

The prevalence of glaucoma is expected to increase. The elderly population is growing, and the prevalence of glaucoma increases with age [3]. Advances in diagnostic technologies also allow for earlier detection [4,5,6]. Furthermore, in the case of the UK, the National Institute for Health and Care Excellence (NICE) released guidance on the diagnosis and management of glaucoma [7], which resulted in an increase in the total number of referrals to hospitals [8,9]. Many hospital eye departments fear capacity problems, since the increase in newly diagnosed cases is not followed by a proportional increase in the number of ophthalmologists [10,11,12]. Ophthalmologists will be obliged to stretch the time intervals between follow-up (FU) appointments, with the risk of not detecting glaucoma progression early on [13]. Moreover, patients with a known higher risk of blindness will be prioritized, thereby reducing access for new patients [14].

One solution is to increase the number of ophthalmologists, which is not feasible in most cases [15]. Another solution is to increase efficiency; studies have identified both shared care (SC) and virtual clinics (VCs) as alternative methods offering a safe, efficient and accepted framework for glaucoma care [16]. For the care of other chronic diseases, such as asthma [17] or diabetes [18], SC schemes have already been demonstrated to be safe, cost-effective and acceptable. VCs have been demonstrated to be beneficial in the care of suspected melanoma [19] and chronic kidney diseases [20].

The aim of this paper was to review the existing literature concerning SC and VCs running in a hospital-based setting. For each scheme, the implementation was investigated, including the role delegation between the different Health Care Providers (HCPs) and the envisioned type of patients. The quality, the productivity and the acceptance of the care delivered were also examined.

## 2. Methods

### 2.1. Study Selection

A literature search was performed using the MEDLINE (PubMed), Embase and Trip databases to identify articles concerning SC and VCs published between January 2000 and July 2021. Relevant abstracts from the annual meeting of the Association for Research in Vision and Ophthalmology (ARVO) of the previous five years were also included.

Two different search queries were made, both using the keywords [glaucoma], [ocular hypertension] and [outpatient clinic]. One search query additionally included the keywords [shared care], [allied health personnel], [collaborative care] and [patient care]. The second search query additionally include the keywords [virtual clinic], [virtual system], and [user computer interface].

Moreover, in order to expand the search, we conducted backward citation tracking, by examining the included article’s reference lists.

### 2.2. Inclusion and Exclusion Criteria

The following inclusion criteria were used: (1) studies evaluating the organization of the glaucoma care pathway; (2) studies evaluating an alternative way of practice, i.e., SC and/or VCs; (3) patients with the diagnosis of glaucoma or suspected glaucoma, or patients at risk of developing glaucoma, e.g., having ocular hypertension (OHT); (4) the staff, working at the clinic, had to be at least one medical Glaucoma Expert (GE) and one non-medical HCP; (5) the clinic operating in a hospital-based setting.

The following exclusion criteria were used: (1) studies evaluating referral patterns from a community-based clinic; (2) studies evaluating case-finding by screening the general population; (3) only optometrists running the clinic; (4) only doctors running the clinic; (5) studies evaluating the care delivered by non-medical HCPs against the care delivered by general ophthalmologists without training or experience in the subspecialty of glaucoma; (6) tele-medicine; (7) virtual reality; (8) the clinic operating in a community-based setting only; (9) the following types of publications (editorials, commentaries, letters); (10) animal studies or in-vitro studies. Only publications in English were considered.

### 2.3. Terminology

#### 2.3.1. Glaucoma Subtypes

One of the most prevalent subtypes is open-angle glaucoma, where the anterior chamber angle is open and the intra-ocular pressure (IOP) is usually elevated, i.e., above 21 mm Hg [21].

Another subtype is closed-angle glaucoma, where the anterior chamber angle is closed. Despite being less prevalent worldwide, closed-angle glaucoma carries a much higher risk of blindness because narrow/closed angles can lead to very high IOP levels in a short period of time [22].

A glaucoma suspect is characterized by having glaucomatous visual field defects or glaucomatous structural optic nerve defects (and not both) [23]. OHT is a condition of having a documented IOP > 21 mm Hg without evidence of visual or structural glaucomatous damage [24].

#### 2.3.2. HCP Working in the Clinic

In this review, ophthalmologists will be referred to as Glaucoma Experts (GEs). No specific criteria for the training were used, since this certification process has only been established in recent years through the collaboration between the European Board of Ophthalmology (EBO) and the European Glaucoma Society (EGS) [25].

The non-medical staff may consist of ophthalmic nurse practitioners (ONP), orthoptists, optometrists and ophthalmic technicians, with different training and responsibilities [26].

Kappa (κ) values were used to measure the chance-corrected agreement between HCPs on a scale of 0.00 to 1.00, indicating no to perfect agreement, respectively. The nomenclature of Landis and Koch was adopted for denominating different kappa ranges as follows: 0.00–0.20 as “slight”, 0.21–0.40 as “fair”, 0.41–060 as “moderate”, 0.61–0.80 as “substantial” and 0.81–1.00 as “almost perfect” agreement [27].

#### 2.3.3. Organization of the Clinic

In standard care (StC) of glaucoma, patients have their appointment with a GE who makes the diagnosis, sets up a management plan during the initial assessment, and decides on a possible change of the management plan during follow-up.

In SC, the non-medical staff assesses patients during most appointments alone. At regular intervals, or earlier if the patient meets the referral criteria, an appointment is planned with the GE, who will examine the patient face-to-face, as is the case in the StC.

In a VC, both HCPs assess the patient at each appointment, with the non-medical staff assessing the patient in a face-to-face consultation and the GE examining the patient remotely, by virtually reviewing the data collected by the non-medical staff.

## 3. Results of Shared Care Studies

### 3.1. Study Selection

From the 400 articles identified, 13 were selected, complemented with one additional article obtained from the reference lists. The processes of identification, screening, duplicates removal and full-text assessment are shown in Figure 1.

### 3.2. Description of the Included Articles

Illustration of the baseline characteristics of the selected articles, including study design, year, location, hospital, SC clinic and population (Table 1).

#### 3.2.1. Recommendations for Shared Glaucoma Care

Two articles provided a model of reference and recommendations on how glaucoma care could benefit from involving the non-medical staff [28,29]. The recommendations of the Australian and New Zealand Glaucoma Interest Group and the Royal Australian and New Zealand College of Ophthalmologists (ANGIG&RANZCO) [28] followed the National Health and Medical Research Council (NHMRC) guidelines [30], and the recommendations from the Canadian Glaucoma Society Committee (CGSC) [29] followed the Canadian Ophthalmology glaucoma clinical practice guidelines (COSgcpg) [31]. For their risk assessment, however, they used the same guidelines [32]. They did not examine the performance of the non-medical staff nor the performance of a SC scheme in general.

#### 3.2.2. Implementation and Performance of SC Clinics

These articles compared the performance of the non-medical staff with that of the GE, or compared SC in general with StC [33,34,35,36,37,38,39,40,41,42,43,44]. All articles concentrated on an actual clinic performing SC, including the glaucoma follow-up unit in the Rotterdam Eye Hospital (SC-GFU) [36,37,38], the Mayo clinic (SC-MC) [33,34,35], the Moorfield Eye Hospital (SC-MEH) [40,42], the Queen’s medical centre (SC-QMC) [39], the Stable Glaucoma Clinic in New Zealand (SC-SGC) [43], the Glaucoma Management Clinic in Australia (SC-GMC) [44] and one established between the Royal Victorian Eye & Ear Hospital and the Australian College of Optometry (SC-RVAC) [41]. The corresponding StC clinic in each hospital is referred as “StC—(name of the corresponding clinic/hospital)”.

### 3.3. The Organization: Implementing SC for Glaucoma

#### 3.3.1. The Role of the GE

As a common rule, new patients had to be assessed by a GE, who decided on their diagnosis, set the target intra-ocular pressure (tIOP) and implemented a management plan. The recommendations of both the ANGIG&RANZCO [28] and the CGSC [29] formed an exception as they considered the GE’s initial assessment unnecessary when a new patient was initially assessed by the non-medical staff and judged to be of low-to-moderate risk [29], without significant ocular risk factors [28]. After the initial assessment, the GE still examined the patient, however less frequently than in StC.

#### 3.3.2. The Role of the Non-Medical Staff

The prerequisite skills of the non-medical staff working in the corresponding SC scheme are listed in Table 2.

In all SC schemes/recommendations, the non-medical staff had to perform and interpret a patient’s history, visual acuity (VA), IOP and visual field (VF). Depending on the SC scheme/recommendation, their required skill set also included optic disc assessment, slit-lamp examination of the posterior segment, assessment of GDx, OCT and HRT, measuring the central corneal thickness (CCT), gonioscopy and/or fundus photography.

#### 3.3.3. Patient Characteristics

Table 3 provides an overview of the characteristics that render patients suitable or unsuitable for each SC clinic, along with a list of conditions requiring referral to a GE.

Unsuitable

As a general rule, new patients were considered unsuitable and needed an initial assessment by a GE. According to Ho et al. [39], decision-making at a first appointment was more related to diagnosis rather than continuing management.

Except for the SC-MEH [40,42] and the SC-QMC [39], patients with unstable glaucoma were considered unsuitable. A patient was considered to be unstable, if the tIOP was exceeded, or if progression was detected using functional or structural testing.

Complicated cases were also excluded, due to the high risk of visual loss. Patients were deemed to fall into this category if they had other eye diseases, advanced glaucoma (definite optic disc pathology or repeatable visual field loss over 12 dB and/or within 10 degrees of fixation, with or without normal IOP [28]), clinical complications or (recently) underwent surgery or laser therapy.

Suitable

Generally, a patient was considered suitable when being stable, a glaucoma suspect, or with a low-to-moderate risk of visual loss.

Back-referral

A patient could be referred back to the GE, in case of patient-specific conditions or at regular intervals, regardless of the glaucoma status, as an internal quality check.

### 3.4. Impact on Glaucoma Care

#### 3.4.1. Quality of Care (QoC)

Distinction is made between the QoC provided by the non-medical staff and the QoC provided by the SC clinic in general (see Table 4). Quality is measured by evaluating completeness, accuracy and management decisions.

##### Performance of the Non-Medical Staff

Performance of the non-medical staff was evaluated by comparison with the “gold standard”, which was the performance of the GE or a working protocol of the corresponding SC clinic.

Completeness of data collection:

The non-medical staff working at the SC-GFU performed the required tests as per protocol in almost all visits [36,37,38]. VF was the only test with a poor compliance, i.e., in only 25.4% of the visits that required VF according to the protocol, the test was actually performed. Of note, also in the StC-GFU run by the GE, VFs were only performed in 16.9% of the visits where a VF was required according to the protocol [36].

Accuracy of data collection:

Agreement between GEs and optometrists on IOP was evaluated in two studies [33,40] and was found to be good. Banes et al. [40] noted that optometrists tended to record lower IOP, but differences were small.

Agreement on structural glaucomatous damage was evaluated in three studies [33,40,42]. When performing slit-lamp examination, the optometrist’s cup/disc ratio was comparable to that of GEs [40] and the optometrist’s ability to decide whether or not an optic disc was glaucomatous was also found to be good (sensitivity and specificity ~83.0%) [42]. When evaluating fundus photographs on stability, the agreement between all HCP (GEs and optometrists) of the SC-MC [33] was found to be good and comparable to the agreement between GEs alone. Banes et al. [40] demonstrated the (dis)agreement rate to be independent of the cup/disc ratio values. Only Shah et al. [33] examined the agreement on OCT interpretation between all HCPs, including GEs and optometrists, and found it to be “fair”. The study of Phu et al. [44] evaluated the agreement between optometrists and ophthalmologists on gonioscopy. The agreement in the exact assessment of the angle was “fair to moderate”. Consistency with a final diagnosis, whether the angle was open or closed, was 93.4% [44].

Agreement on functional glaucomatous damage was evaluated in four studies [33,39,40,42]. The agreement on VF-status was “fair” [42], “moderate” [33] and “almost perfect” [39,40]. Banes et al. [42] pointed out that optometrists were more cautious than GEs, by classifying more eyes as being “progressive”.

Management decisions:

Several studies examined the non-medical staff’s ability to make management decisions based on their interpretation of tests and examinations [33,36,37,39,40,42].

As for glaucoma status, in the SC-GFU, the non-medical staff referred half of the cases which met one of the back-referral criteria in the protocol back to the GE. Out of the cases that met the GDx- or VF-criterion (indicating suspected progressive damage), 92.0% and 75.0% of the cases, respectively, were actually sent back. These values amounted to 66.7% for the IOP-criterium (IOP > tIOP) and 36.0% for the VA-criterium (declined ≥ 2 lines) [37]. In the SC-GFU, the non-medical staff could also opt to seek advice of the GE when one the above criteria was met. In 100% of the cases that met the GDx- or the VF-criterium, the non-medical staff asked for advice or referred back. This value amounted to 84.6% and 68.2% for the IOP- and VA-criterium, respectively [36].

In the SC-MC, disease progression was defined as IOP > tIOP, progression on optic nerve photographs, OCT or VF [33,34]. Shah et al. [33] showed a “fair” level of agreement on glaucoma progression diagnosis between all HCPs (optometrists and GEs) and between GEs alone. The level of agreement between all HCPs was higher when relying on IOP or disc hemorrhages compared with the agreement when relying on OCT or VF [33]. Of all the available test data, the OCT and VF data were considerably less used by the optometrists than by the GEs. This discrepancy in use was also reflected in the high discrepancy in interpreting OCT and VF between all HCPs (agreement of 36.0%, κ = 0.26, for OCT, and agreement of 53%, κ = 0.45, for VF) [33].

Two other articles evaluated the agreement between HCPs on whether a patient should be discussed with the GE [39,42]. Ho et al. [39] found this agreement to be “almost perfect” between the GE and the non-medical staff. Banes et al. [42] found this agreement to be slightly smaller (72.0%), but equal to the agreement between two GEs on whether a patient should be discussed with them. Three studies [39,40,42] evaluated the agreement on disease status by using the proposed follow-up interval as a measure. A shortened interval indicated that the disease status was judged to be worsening. Overall, the agreement was “almost perfect” [39,40]. Only Banes et al. [42] showed a “fair” agreement.

As for ordering tests, the non-medical staff of the SC-MH [40,42] and the SC-QMC [39] was allowed to do so. Ho et al. [39] showed a high agreement on ordering a VF at the next appointment between the optometrists and GE. Although Banes et al. [42] assessed lower values, the agreement was still good and similar to the agreement between two GEs. In both clinics, the optometrists tended to order more additional tests than the GEs [39,40,42].

The non-medical staff of the SC-MH [40,42] and the SC-QMC [39] were also able to decide on further treatment. In both clinics, agreement was high for both the medical and surgical treatments.

##### Performance of the SC Clinic

In this case, the “gold standard” corresponds to the StC or the guidelines used (Table 4).

Completeness of data collection:

The Mayo Clinic showed an increase in compliance on initial testing to the American Academy of Ophthalmology Preferred Practice Pattern (AAO PPP) guidelines [45] after implementation of the SC-MC [35]. Similarly, the SC-RVAC [41] showed a high compliance to both the AAO PPP [45] and the ANGIG&RANZCO recommendations [28]. The compliance on rate of testing was weak, but similar, for VF in both SC-GFU and StC-GFU [36,37].

Accuracy of data collection:

No difference was found between the results obtained by the SC-GFU [36,37] and the StC-GFU. One exception was VA, which declined in more visits of the StC-GFU than in the SC-GFU [36]. Holtzer-Goor et al. attributed this difference to the different protocols used in both clinics; the SC-GFU had to perform VA at every visit while the StC-GFU had to perform VA only when judged to be necessary (but at least once a year) [36]. In other words, the StC-GFU would mainly perform VA for those patients who mentioned having difficulties with their sight [36]. The implementation of SC-RVAC resulted in a 14.0% increase in correct diagnosis when assessing the optic nerve compared to the StC clinic [41].

Management decisions:

No difference was found between the StC-GFU and the SC-GFU in the decision on the number of patients judged to be stable or progressive [36,38]. Holtzer-Goor et al. concluded that a SC scheme did not miss a significant number of cases of suspected progression [36]. Damento et al. assessed the decision on “disease status” in the Mayo Clinic by using the “number of patient visits” as a measure [34]. The rationale was that, if an HCP judged the disease status to be worsening, that HCP decided to shorten the follow-up interval, which resulted in more visits taking place in a certain amount of time. No difference was found in the number of patient visits between the SC-MC and the StC-MC [34].

Furthermore, the number of treatment changes was similar between the SC-GFU and the StC-GFU [36,37,38]. Moreover, no difference was found concerning the reason for change, i.e., IOP exceeding the tIOP, intolerance to the medication, structural or functional progression [36,37]. Likewise, the number of procedures carried out in the SC-MC and the StC-MC did not differ [34]. However, the number of procedures performed by the GE tended to increase after implementation of the SC scheme [34].

#### 3.4.2. Acceptance

##### Patients

Patient satisfaction was about the same in the SC-GFU and the StC-GFU [36,37]. No difference was noted in the dimensions “overall mark”, “knowledge”, “waiting area”, and “information received”. Patients scored the SC-GFU higher on ‘taking sufficient time” and “giving sufficient information”.

When comparing HCPs, Holtzer-Goor et al. assessed a higher score on the “overall mark” for the non-medical staff [36]. The GE got a higher score on the dimension “information received”. Patients gave both the GE and the non-medical staff similar scores on “knowledge” and “waiting area”. Lemij et al. [37] found similar scores as Holtzer-Goor et al. [36], but assessed a higher score for the GE on “knowledge” and “information received”. In the SC-RVAC, almost 95% of the responders opted to be treated in the SC-RVAC rather than remaining on the waiting list of the StC-RVAC [41].

##### Staff

All clinicians of the SC-RVAC found the SC clinic an excellent opportunity to exchange knowledge, and 82.0% wanted to stay working in the clinic [41]. Similarly, the GE and the ophthalmic technicians were very pleased to work in the SC-GFU [37,38]. The ophthalmic technicians indicated the patient contact and the increased responsibility to be the main reasons. However, the optometrists working in the SC-GFU found their work tedious, and thought the shared clinic was not working satisfactorily [37].

#### 3.4.3. Productivity

In the SC-RVAC, the waiting list was reduced by 32.0% after 17 months and by 92.0% after 28 months [41]. Holtzer-Goor et al. hypothesized that the implementation of the GFU reduced the waiting list, because of the increased number of patients (+23.0%) and patient visits (+16.0%) [38]. Another article on the SC-GFU by Holtzer-Goor et al. showed that for each patient transferred to the SC-GFU, approximately 0.57 extra stable glaucoma patients could be managed in the hospital [36]. However, this seemed to be a short-term effect. In the long term, the patients’ outflow would be limited because glaucoma is a chronic disease [36]. Moreover, the inflow would increase as the number of patients with glaucoma is predicted to increase as indicated above (cfr. Section 1). Damento et al. documented an increased access for complex patients to the GE after implementing the SC-MC [34]. Botha et al. demonstrated an improvement in IOP control and decreased progression rates since the implementation of SC-SGC, partly attributable to less delays in follow-up [43].

## 4. Results of Virtual Clinics’ Studies

### 4.1. Literature Search

From the 445 articles identified, 14 were selected, complemented with one additional article obtained from the reference lists. The processes of identification, screening, duplicates removal and full-text assessment are shown in Figure 2.

### 4.2. Description of the Included Articles

Baseline characteristics of the included articles are shown in Table 5. Unlike SC clinics, VCs are not widespread. All VCs included in this review are located in the UK. A VC could be implemented in the initial assessment of a patient who had been referred from primary care, in which case these clinics served as a triaging service. These types of VCs included the glaucoma assessment clinic (GAC) [46,47,48] at the Singleton hospital and the Glaucoma Screening clinic (GSC) [49] as part of a broader service transformation program being established at the MEH. Gunn et al. did not focus on a specific VC but investigated the proportion, the characteristics and the acceptability of the Hospital Eye Service (HES)-units that implemented a VC for glaucoma care [50].

A VC could also play a role in patient follow-up. These types of VCs included the virtual triaging clinic established in Bristol, Nuneaton and Kingston, referred to as the glaucoma classifying clinic (GCC) [51] in this paper, the virtual clinic in Princess Alexandra Eye Pavilion (VC-PAEP) [52,53], the virtual clinic in Manchester Royal Eye Hospital (VC-MREH) [54] and in Bristol Eye Hospital (VC-BREH) [54], the virtual clinic in The Royal Eye Infirmary Plymouth (VC-REIP) [55] and the Stable Monitoring Service [56,57,58] (SMS) which was the other part of the virtual service transformation implemented at the MEH [50]. The complete virtual service at MEH was named the Glaucoma Screening and Stable Monitoring Service (GSMS) [59], implementing both the GSC [49] for initial assessment and the SMS [56,57,58] for patient follow-up. Nikita at al studied expanded patient eligibility criteria at MEH (VC-MEH); both new and follow-up patients were included [60].

### 4.3. The Organization: Implementing a VC for Glaucoma

#### 4.3.1. Role of the Staff

The prerequisite skills of the non-medical staff working in the corresponding VC are listed in Table 6.

#### 4.3.2. Initial Assessment

In the GSC, a new patient was evaluated by a clinician who decided if the patient was eligible for the VC, based on the referral letter from primary care [49]. If eligible, the patient underwent testing, performed by ophthalmic technicians. Patients were also given a questionnaire enquiring about their medical and family history. The GE reviewed the collected clinical data and decided on a follow-up at the hospital or a discharge. In the GAC, new patients were systematically seen by the ONP without prior triage [46,47,48]. The ONP took the patient’s history, performed tests, clinically examined the patient and assessed the patient’s risk profile. The article of Choong et al. could be considered as the pilot study of the GAC, which did not include a VC yet [47]. It was presented as a fast-track system, which allowed an ONP to triage patients, including defining the time-interval in which patients needed to have their face-to-face appointment with the GE. Rathod et al. built further on this system and added a virtual service [48]. The ONP would again triage these patients, but the GE would do the initial assessment by reviewing the GAC data instead of a face-to-face assessment [48].

#### 4.3.3. Follow-Up

The non-medical staff working at SMS varied between articles [56,57,58]. In the pilot study, ophthalmic technicians performed VA, VF and optic disc imaging [56]. An ONP took the patient’s history by reviewing a questionnaire and was in charge of the clinical examination, including tests that required more expertise (IOP, slit-lamp examination). Furthermore, they could offer advice on common eye complaints and drop delivery technique. The GE reviewed the notes of previous appointments and included these in the management decisions. In a later study, the ONP was removed from the SMS [57]. A slit-lamp examination was no longer performed, and the ophthalmic technicians took over IOP measurement. In the study of Nikita et al., the SMS also incorporated OCT for the virtual review [58]. However, the profession of the non-medical staff was not specified in this article [58]. Similar to the GSC [49], a follow-up patient was evaluated by a clinician who decided if the patient was eligible for the SMS [59]. In the GCC, an optometrist supported by ophthalmic technicians met the patient first and collected clinical data from the clinical history, clinical examination, VF and color optic disc images [51]. After data interpretation, the optometrist classified the patient into one of five risk categories, each associated with a required time interval for a face-to-face consultation with a GE. Subsequently, in the virtual review, the GE confirmed or changed the optometrist’s classification, based on the same clinical data. If a patient was classified to have no strong evidence of glaucoma, the patient would be discharged, and would not be seen by a GE [51].

#### 4.3.4. Patient Suitability

Table 7 provides an overview of new patients who are considered suitable and unsuitable for each VC.

##### New Patients

Suitable:

At the start of the GSC, only “low risk” glaucoma suspects were considered to be suitable [49]. These patients only had one of the following risk factors: suspicious optic discs, suspicious VF or IOP >20 mm Hg. In a second stage, “low-to-moderate risk” glaucoma suspects (having up to two risk factors, including and a positive family history in a first-degree relative) were also eligible for the GSC [49]. Finally, at a later stage, patients could have up to three of those risk factors and still be eligible for the GSC [49]. A clinician decided if a patient would be included in the VC or would be sent to the GE immediately. In the GAC, all new glaucoma/OHT suspect referrals were included in the VC [46,48].

Unsuitable:

The clinician excluded new glaucoma/OHT suspect referrals from the GSC if they did not meet the inclusion criteria [49]. Patients were also excluded if they showed definitive signs of glaucoma, were angle-closure suspects or were referred with an IOP >32 mm Hg [49]. If a patient showed an IOP >32 mm Hg at the initial assessment of the GSC, the patient would be sent to a GE on the same day [49].

##### Follow-Up Patients

Suitable:

Only stable patients with a low risk of glaucomatous damage progression were suitable for the SMS [56,57,58]. In the “pilot” study by Clarke et al., patients were included if their planned follow-up frequency was more than six months [56]. This study concluded that patients were suitable if they had stable glaucoma and were at low risk of progression to significant visual loss over each follow-up interval [56]. Recent studies conducted by Nikita et al. expanded patient suitability from glaucoma suspects and low-risk glaucoma to most types of glaucoma and in various stages of disease progression, and provided firm evidence that expanded patient eligibility criteria are able to deliver high-quality glaucoma care that is safe and effective [58,60]. In the GCC, patients were taken from the general follow-up pool and could have any type of glaucoma at any stage [51]. In the VC-PAEP, patients with mild to moderate stable open-angle glaucoma or patients with mild to moderate stable primary closed-angle glaucoma who are bilaterally pseudophakic were suitable [52,53].

Unsuitable:

In the “pilot” study of the SMS, patients were excluded if they had poor mobility or if the quality of their VF or fundus photographs was poor [56]. When the SMS was eventually established, other exclusion criteria were added, including monocular, concurrent eye diseases/morbidities, a low VA and if there were concerns about the patients’ adherence to treatment [57]. The expanded monitoring service studied by Nikita et al. did not specify the exclusion criteria [58]. Both the SMS and the VC-PAEP excluded phakic angle-closure glaucoma/glaucoma suspects and patients who had a history of glaucoma filtration surgery [52,53,57].

### 4.4. Impact on Glaucoma Care

#### 4.4.1. QoC

##### New Patients (GSC and GAC)

In the GSC, 20.0% of patients were discharged wrongly by the GE, but only a minority required medical intervention, leading to a “significant” false rate of 4.0% [49]. The GSC missed two narrow angles with one requiring surgery [49]. In the GAC, the similarity of a GE’s virtual assessment was “substantial” (κ = 0.72) to those made through a face-to-face assessment [48].

##### Follow-Up Patients (GCC, SMS, VC-PAEP, VC-MREH and VC-BEH, VC-REIP)

In the GCC, a “substantial” (κ = 0.69) agreement on triaging was found between the optometrists and supervising GEs [51]. In general, optometrists tended to be overcautious by considering patients more at risk. Still, the optometrists discharged 15.0% of the cases having glaucoma according to the GE. Another concern was the 6.5% of cases considered as low-risk by the optometrist who were identified as unstable by the GE [51]. Kotecha et al. compared the face-to-face assessment by a GE in the SMS with a virtual assessment by a different GE (inter-GE agreement) or by the same GE (intra-GE agreement) [57]. The inter-GE agreement was found to be “fair” (κ = 0.32). In this analysis, seven out of 14 unstable cases were detected during the virtual review (sensitivity of 50.0%). The other seven patients (3.4% of all patients) had been “misclassified” as stable during the virtual clinic assessment, two of whom (1.9%) having advanced VF loss. The sensitivity increased to 75.0% when only considering consultants and excluding fellows from the GE population [57]. Regarding the analysis made by the same GE, the intra-GE agreement was “fair” (κ = 0.26–0.27). The disagreements would only pose a risk for six patients (3.1% of all patients), since these were deemed as stable during the virtual review, but unstable at the face-to-face review by the same GE. The sensitivity amounted to 75.0% for the consultant and 60.0% for the fellow [57]. The study of Mostafa et al. showed that Goldmann applanation tonometry measurements only have moderate agreement when performed by different operators and that repeat Ocular Response Analyzer (ORA) IOP measurements were more consistent [53].

#### 4.4.2. Acceptance

##### Patients

According to Kotecha et al., patients with a low risk of progression were more open to a VC [59]. Patients were pleased with the reduced waiting time, the expertise of the staff and the productivity of the VC [57,59]. Court and Austin found that patients in the VC did not consider they were receiving inferior quality care compared to patients in StC [46]. However, some patients were disappointed by not receiving immediate feedback and not seeing a doctor on the same day [57]. Tatham et al. found no significant difference in knowledge of glaucoma between patients of VC-PAEP and StC-PAEP, suggesting that patients’ knowledge is not disadvantaged by virtual clinics [52]. Study patients of Gunn et al. reported reduced waiting times as a key aspect of positive experiences [54]. These patients demonstrated high levels of trust in the staff performing tests in the glaucoma VC [53]. Spackman et al. evaluated patient satisfaction with the glaucoma VC in comparison with StC in The Royal Eye Infirmary Plymouth [55]. Overall, 98% of patients felt that the VC was the same or better than the StC [55].

##### Staff

Gunn et al. [50] investigated the perspective of the GE; 92.9% of the respondents considered the VC as safe and efficient as StC, with 31.0% rating the efficiency as very good. The authors also identified the main reasons for not implementing a VC: insufficient staff, inadequate space, insufficient time or funding to train the non-medical staff, the risk of missing pathology and the lack of face-to-face discussion [50]. Later, Gunn et al. [54] investigated the perceptions of the technicians working in the glaucoma SC clinic. The technicians reported satisfaction in working within the glaucoma service. However, they commonly felt they would benefit from more detailed training, particularly around knowledge of the conditions and medications [54].

#### 4.4.3. Productivity

In the GSC, the GE discharged 62.0% of new glaucoma/OHT suspect referrals, sent 1.0% for an urgent same-day assessment with a GE, referred 6.0% to SMS and booked 31.0% for the consultant-led outpatient clinic [49]. In the GAC, 20.5% were discharged, after being diagnosed virtually as “normal” [48]. In the GCC, the GE discharged 3.7% of new glaucoma/OHT suspect referrals, which is 1.2% more than the number of patients that would have been discharged by the optometrists [51]. The virtual supervision by the GE also reduced the number of additional visits, e.g., the follow-up appointments, by 2.4% of the total number of visits [51]. The implementation of the SMS led to 13.0% of the patients being discharged, 57.0% of the patients being rebooked in the SMS and 30.0% being sent to a GE for a face-to-face appointment [58].

## 5. Discussion

### 5.1. Application of SC/VC

All the studies regarding SC clinics in this review concentrated on patient follow-up, while the studies regarding VCs were done with a follow up setting—GCC [51], SMS [56,57,59], VC-PAEP [52,53], VC-MREH [54], VC-BEH [54] and VC-REIP [55] or for an initial assessment only—GAC [46,47,48] and GSC [49].

### 5.2. Generalizability to Other Hospitals/Countries

The guidelines for the management of glaucoma are mainly country-specific: the AAO PPP [45] in the USA, the Royal College of Ophthalmologists’ (RCO) guidelines [61] in the UK, the COSgcpg [31] in Canada and the NHMRC guidelines [30] in Australia. The guidelines from the RCO [61] and guidelines from NICE [62] are commonly used in the UK, however not in other countries. Furthermore, in the VCs of the UK-based articles, the non-medical staff entrusted with a particular task probably followed the same UK-based required training. Hence, one should be careful in extrapolating to other countries, e.g., the non-medical staff from the REH are not trained to perform slit-lamp examination to assess the optic disc [36].

### 5.3. Skills of the Non-Medical Staff

In all SC clinics and VCs operating during follow-up, the non-medical staff had to take a clinical history, measure IOP and perform a functional (VF) and a structural (fundus photographs, OCT, HRT or GDx) evaluation. In all SC clinics [33,34,35,36,37,38,39,40,41,42,43,44], at least one non-medical staff member had to interpret the results from these examinations to decide on the glaucoma status and the (possible) presence of progression. In the VCs [46,47,48,49,50,51,56,57,58,59] the non-medical staff had to perform all examinations a GE would normally do without making any treatment decisions. In only two VCs (GCC and GAC), a non-medical staff member had to be able to interpret these examinations to triage patients [46,48,51]. In the other virtual services, (GSMS, VC-PAEP, VC-MREH, VC-BEH, VC-MEH and VC-REIP), the non-medical staff had only to collect and to deliver data to the GE.

### 5.4. Suitable Patients

In all clinics, patients who were stable and were at low risk of progression were considered suitable. Patients with narrow angles, with or without glaucoma, were found suitable, if the non-medical staff was able to assess the angle of the anterior chamber; hence, such patients were only accepted in the GCC, GAC, SC-MEH and the SC-QMC [41,42,43,48,51].

### 5.5. Pathway of a High-Risk Patient

In the GSMS, a clinician triaged both new patients and follow-up patients, whereby high-risk patients were sent directly to the GE for a face-to-face appointment [49,56,57,58,59]. The GAC and the GCC, however, did not foresee such triage system, in that the non-medical staff assessed all new patients [48,51]. However, high-risk patients were sent to a GE immediately.

In all implemented SC schemes, a GE assessed all new patients to decide on their eligibility. The ANGIG&RANZCO [28] and the Canadian Glaucoma Society [29] recommendations on SC were an exception in that the initial assessment by a GE was not mandatory if the non-medical staff considered a new patient to be low-to-moderate risk [29], without significant ocular risk factors [28].

### 5.6. Compliance to Guidelines

An increase in compliance with guidelines was noted when implementing a SC clinic due to the combined examination efforts of the non-medical staff and the GE [35,41,47]. Compliance was also higher when following a standardized protocol [35,36,47]. Moreover, by delegating some tasks to the non-medical staff, the GE would have more time and would not have to give up examinations [40]. Banes et al. showed that the lack of time in the often very busy StC clinic caused the GE to skip some examinations [40]. Such was also the case in the GFU, where a low compliance rate was noted in the SC-GFU as well as in the StC-GFU [36,62]. This clinic admitted only low risk patients with no proven glaucoma (but with positive family history, OHT and/or suspicious looking discs) or early glaucoma damage. In such cases, structural measurements were deemed to be more important, for being more informative and also quicker to perform than VFs [36,62]. None of the VC-articles examined the effect on compliance.

### 5.7. Data Interpretation: Importance of Training

The accuracy of the data interpretation increased with the level of experience/training of the non-medical staff. As the optometrists from the SC-MEH [40,42] and SC-QMC [39] got extra training in these tasks, their interpretation of the fundus photographs and VF was more accurate than in the SC-MC. The lack of training could also explain why the agreement on evaluating OCT between optometrists from the SC-MC and GEs was less good, and worse than the agreement between GEs [33]. The optometrists from the SC-MEH and SC-QMC assessed the optic disc through slit-lamp examination and showed a high agreement with the GE because they were trained to use these devices [39,40,42]. None of the VC-articles investigated the accuracy of the non-medical staff.

### 5.8. Quality of Management Decisions

In their decisions on progression/referral, the non-medical staff of the SC-GFU followed the referral criteria strictly [36,37]. More importantly, adherence to these criteria increased when, besides referring patients to the GE directly [37], they could also ask for the GE’s advice [36]. The level of agreement on progression between optometrists and the GEs from the SC-MC was only “fair”, but similar to the level of agreement between two GEs. A point of concern was that almost 1/3 of the glaucoma cases being progressive would not be referred to the GE. Possible reasons were an incorrect interpretation of data (see above) and not using all data when making decisions. However, optometrists tended to be overcautious in general. In both the SC-MEH and GCC, the optometrists classified more patients as progressive or at higher risk than the GE [42,51]. Most likely, the reason was to make a safer decision. As a consequence, the optometrists tended to order more additional tests than the GEs [39,40,42].

Decisions on discharge/follow-up were also safeguarded. In the GCC, decisions of the non-medical staff in this respect were supervised virtually by the GE [51]. In the GFU, the non-medical staff could not discharge and could only decide to keep or shorten the interval as planned [36,37,38]. Similarly, in the GAC the non-medical staff could not discharge, and the GE would (virtually) assess the patient within a maximum time interval of three months [48]. The agreement with the face-to-face diagnosis was also high. In the GSC out of the 16 patients for whom the diagnosis differed between the face-to-face and virtual review, only three patients required medical intervention [49]. Two of these patients were diagnosed as having OHT, one of whom had an IOP at the face-to-face consultation which was twice as high as what had been found in the virtual review and in the referral letter. The third patient had narrow, occludable angles requiring prophylactic laser iridotomy [49]. In the SMS, the sensitivity of detecting unstable cases was dependent on the expertise of the GE; a higher sensitivity was noted for the consultant than for the fellow, both in the inter- and intra-observer agreement analyses [56]. The arbitrary stable/unstable classification, which was used in the SMS for deciding on the time to the next FU appointment, was explained as a possible reason for the low sensitivity [56]. However, since only stable and low-to-moderate risk patients could enter the clinic, the actual number of missed unstable patients was low, and even lower with advanced VF loss [56]. Due to the wide confidence intervals, it was suggested to perform more extensive studies to provide a more accurate answer on the virtual clinic’s sensitivity [56]. In the SC-MEH studies, the optometrist(s) and the GE independently decided on the follow-up interval and the further treatment [40,42]. The agreement was high and comparable to the agreement between two GEs. In SC-QMC study, where the GE had to state (dis)approval with the optometrists’ decisions, the agreement was even higher [39].

### 5.9. Acceptance

All findings showed that the acceptance of care provided by a SC clinic or a VC was at least as good as in the StC clinic. In the SC-GFU, the GE scored higher on “knowledge” and “information received”; however, the difference was too small to be relevant [37]. In the study of Spackman et al., 98% of patients felt that the glaucoma VC was the same or better than the StC [55].

Acceptance of SC and VCs by the GEs was overall good; some medical staff however found their work in SC to be tedious [37].

### 5.10. Productivity

The waiting list for new glaucoma/OHT suspect referrals to the GE decreased; most of the follow-up appointments of stable, low risk glaucoma suspects/patients were given to the non-medical staff, thereby saving the GE time [36,41]. Also, the non-medical staff could ensure these patients got their appointments on time.

The access of complex patients and unstable patients to the GE increased [34]. The non-medical staff was made responsible for monitoring stable glaucoma, thereby saving time for the GE to accept more complex patients and to see all patients on time and detect progression quickly. Holtzer-Goor et al. found a significantly lower VA in the StC-GFU; this could indicate that more complex patients were directed to the StC-GFU, thereby achieving one of the main goals of SC [36]. Likewise, the number of procedures performed by the GE tended to increase when cooperating with the non-medical staff in the SC-MC suggesting better access of complex patients to care provided by the GE [34].

By delegating triaging, GEs were also less busy with the initial assessment. The GAC and the GSC respectively sent only 79.5% and 32.0% to the GE for a face-to-face assessment [48,49]. The GSC sent less people to the StC outpatient clinic because they could refer stable OHT/glaucoma suspects/glaucoma patients with low-to-moderate risk to the SMS [49,56,57].

### 5.11. Directions for Future Research

Since hospitals do not always employ all non-medical staff professions, the effect of replacing one profession by another should be studied. Furthermore, the impact of VCs on compliance to guidelines/protocol should be investigated. Decisions made through virtual review were not completely similar to those made through face-to-face assessment, which could be caused by not assessing the patient face-to-face, or by the non-medical staff not providing accurate data. A deeper analysis is needed to improve our knowledge regarding these findings. An economic analysis of SC/VCs versus StC, the long-term effect of SC/VCs on the disease itself and possible synergetic effects when combining SC and VCs are other interesting topics for future studies. Furthermore, since all the VCs in this review are located in the UK, the conclusions drawn may not apply in other countries, especially outside the Anglo-Saxon world. Therefore, future studies conducted outside the UK/Anglo-Saxon world can be an added value.

## 6. Conclusions

This literature review examines different implementations of SC and VCs in a hospital-based setting and compares them with the conventional ophthalmologist-led outpatient service in terms of the QoC delivered, the acceptance and the productivity.

A high acceptance seems to be linked to the reduced waiting time in the clinic and the social skills of some non-medical staff members having contact with the patient. Furthermore, by dividing the workload among the ophthalmologists and the non-medical staff, more patients could enter the glaucoma care pathway and be seen on time. Due to their reduced workload, ophthalmologists could assess new and high-risk patients more rapidly and with access to more auxiliary tests. Progressive glaucoma could be detected earlier, the treatment could be adjusted faster and further damage could be prevented.

In summary, SC and VCs are two promising approaches to tackle the upcoming capacity problems of glaucoma care within a hospital-based setting, without compromising the acceptance and the QoC delivered.

## Figures and Tables

**Figure 1 jcm-10-04785-f001:**
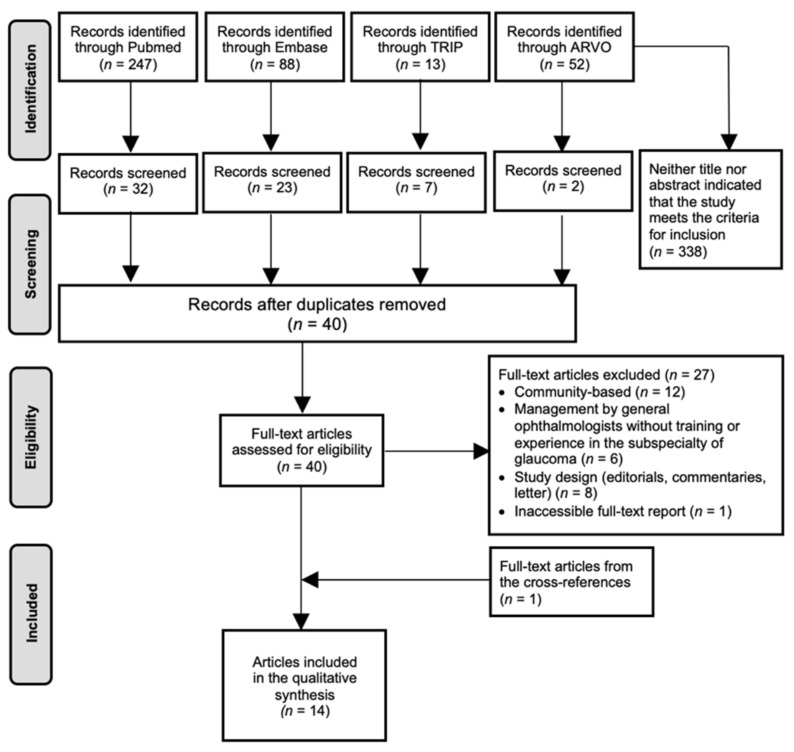
Study selection PRISMA flow chart on Shared Care. Abbreviations: ARVO = Annual Meeting of the Association for Research in Vision and Ophthalmology; *n* = amount.

**Figure 2 jcm-10-04785-f002:**
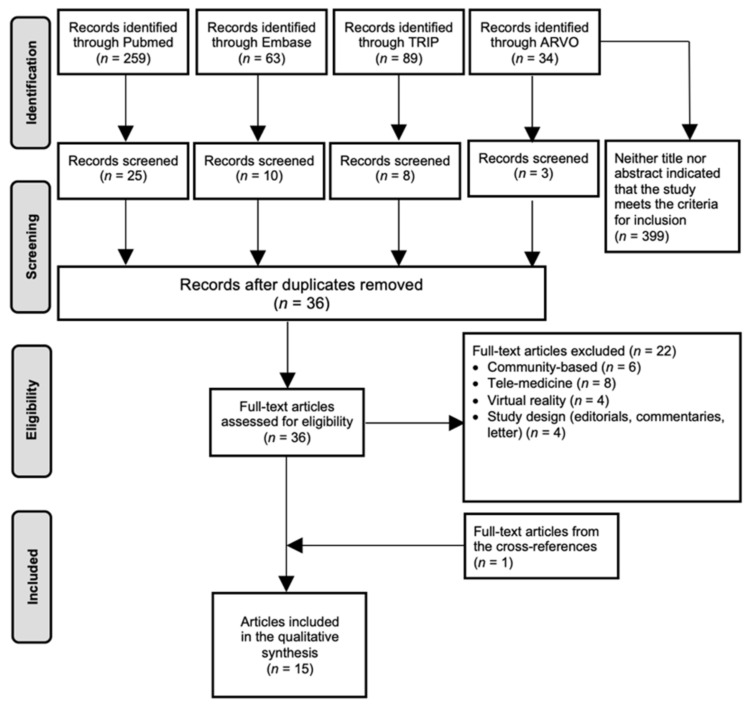
Study selection PRISMA flow chart on Virtual Clinics. Abbreviations: ARVO = Annual Meeting of the Association for Research in Vision and Ophthalmology; *n* = amount.

**Table 1 jcm-10-04785-t001:** Baseline Characteristics “Shared Care”.

First Author, Year	Country	Hospital	SC/Recommendation	Study Sample	NG/OSR vs. FU-Patients
White et al., 2014	Australia and New Zealand	/	ANGIG& RANZCO	/	FU-patients
Bentley et al., 2019	Australia	RVEEH	SC-RVAC	1024 patients	FU-patients
Canadian Glaucoma Society Committee, 2011	Canada	/	CGSC	/	FU-patients
Holtzer-Goor et al., 2010	The Netherlands	REH	CS-GFU	815 patients (2100 visits) SC-GFU: 405 (1181 visits) StC-GFU: 410 (919 visits)	FU-patients
Holtzer-Goor et al., 2016	The Netherlands	REH	CS-GFU	815 patients (2100 visits) SC-GFU: 405 (1181 visits) StC-GFU: 410 (919 visits)	FU-patients
Lemij et al., 2010	The Netherlands	REH	CS-GFU	815 patients (2100 visits) SC-GFU: 405 (1181 visits) StC-GFU: 410 (919 visits)	FU-patients
Damento et al.,2018	USA	MC	SC-MC	358 patients	FU-patients
Winkler et al.,2017	USA	MC	SC-MC	591 patients	FU-patients
Shah et al., 2018	USA	MC	SC-MC	200 patients (299 eyes)	FU-patients
Banes et al., 2000	UK	MEH	SC-MEH	54 patients (102 eyes)	FU-patients
Banes et al., 2006	UK	MEH	SC-MEH	349 patients	FU-patients
Ho et al., 2011	UK	QMC	SC-QMC	140 patients	FU-patients
Bhota et al., 2019	New Zealand	SGC	SC-SGC	509 patients (760 visits)	FU-patients
Phu et al., 2019	Australia	GMC	SC-GMC	101 patients	FU-patients

Abbreviations: SC = shared care clinic; StC = standard care clinic; FU = follow-up; NG/OSR = new glaucoma/ocular hypertension suspect referrals; ANGIG&RANZCO = recommendations of the Australian and New Zealand Glaucoma Interest Group and the Royal Australian and New Zealand College of Ophthalmologists; SC-RVAC = shared care clinic, established between the Royal Victorian Eye & Ear Hospital and the Australian College of Optometry; CGSC = recommendations from the Canadian Glaucoma Society Committee; RVEEH = Royal Victorian Eye & Ear Hospital; GFU = Glaucoma Follow-up Unit; MC = Mayo Clinic’s campus in Rochester; MEH = Moorfield Eye Hospital; QMC = Queen’s medical center; REH = Rotterdam Eye Hospital; USA = United States of America; UK = United Kingdom, SGC = Stable Glaucoma Clinic, GMC = Glaucoma Management Clinic; WEI = Wilmer Eye Institute.

**Table 2 jcm-10-04785-t002:** The prerequisite skills of the non-medical staff working in the corresponding shared care scheme.

SC/ Recommendations	NMS	History Taking	IOP	VA	Slit-Lamp Examination	+Gonio	VF	Fundus Photographs	OCT	HRT	GDx	CCT
ANGIG& RANZCO	NS	x	x	x	x(a&p)	x	x	x	x*	x*	x*	x
SC-RVAC	Opto	NS°	NS°	NS°	NS°	NS°	NS°	NS°	NS°	NS°	NS°	NS°
CGSC	Opto	x	x	NS	x(a&p)	x	x	x	x**	x**	x**	0
SC-GFU	Opto OT	x	x	x	0	0	x	0	0	0	x	0
SC-MC	Opto	x	x	x	x(a&p)	0	x	x	x	0	0	0
SC-MEH	Opto	x	x	NS	x(a&p)	x	x	x***	x***	x***	x***	x***
SC-QMC	Opto	x	x	NS	x(a&p)	x	x	0	x	x	0	x
SC-SGC	Opto	x	x	NS	NS	NS	x	x	0	x	0	NS
SC-GMC	Opto	x	x	x	x(a&p)	x	x	NS	0	x	0	0

Abbreviations: x = task performed by the corresponding member of the non-medical staff; 0 = task not performed by any member of the non-medical staff; x* = preferable rather than mandatory; x** = automated imaging tests, not further specified; x*** = could decide on further assessment if indicated; NS = not specified; NS° = not specified, but based on the recommendations of the Australian and New Zealand Glaucoma Interest Group and the Royal Australian and New Zealand College of Ophthalmologists; Opto = optometrists; OT = Ophthalmic technician; (a&p) = anterior&posterior segment; SC = shared care clinic; NMS = non-medical staff; IOP = intra-ocular pressure; VA = visual acuity; Gonio = gonioscopy; VF = visual field; OCT = optical coherence tomography; HRT = Heidelberg retinal tomography; GDx = GDx ECC scanning laser polarimetry; CCT = central corneal thickness; ANGIG&RANZCO = recommendations of the Australian and New Zealand Glaucoma Interest Group and the Royal Australian and New Zealand College of Ophthalmologists; SC-RVAC = shared care clinic, established between the Royal Victorian Eye & Ear Hospital and the Australian College of Optometry; CGSC = recommendations from the Canadian Glaucoma Society Committee; GFU = Glaucoma Follow-up Unit; MC = Mayo Clinic’s campus in Rochester; MEH = Moorfield Eye Hospital; QMC = Queen’s medical centre; SGC = Stable Glaucoma Clinic in New Zealand; GMC = Glaucoma Management Clinic in Australia.

**Table 3 jcm-10-04785-t003:** The characteristics that render patients suitable or unsuitable for each shared care clinic, along with a list of conditions requiring referral to a Glaucoma Expert.

SC/Recommendations	NMS	Suitable	Unsuitable	Model-Specific Referral	Patient-Specific Referral
ANGIG& RANZCO	NS	OHT; GS; G: stable & low/moderate risk	High risk of visual loss, e.g., other ocular diseases; advanced glaucoma (both stable and unstable); closed angles	GS: every 3–4 y; G early/moderate stable: every 2 y	Recent diagnosis; start of therapy; unstable disease; acutely raised or very high IOP; narrow angles
SC-RVAC	Opto	NS°	NS°	NS°	NS°
CGSC	Opto	OHT; GS; G: stable & low risk; Other concurrent eye diseases related to G	G: unstable/moderate/advanced	GS: every 3–4 y; G early: every 2–3 y	Recent diagnosis; start of therapy; GS with high risk (suspected progression); unstable G; acutely raised or very high IOP
SC-GFU	Opto OT	OHT; GS; G: stable	Complex cases: other ocular diseases; H laser therapy for DRP	G and GS: every third visit	Recent diagnosis; suspected progression
SC-MC	Opto	G: stable (mild/moderate/advanced); Other concurrent eye diseases	G: unstable	Mild G: every 3 y; moderate G: every 2 y; advanced G: every 1 y	Recent diagnosis; suspected progression; significant cataract; intolerant of medications
SC-MEH	Opto	OHT; GS; G; Other concurrent eye diseases	Known clinical complication, H laser therapy/surgery	OHT: every 1 y; stable G: every 6 mo; after change in therapy: 1 mo	Recent diagnosis; changes in treatment
SC-QMC	Opto	OHT; GS; G; Other concurrent eye diseases	H laser therapy/surgery	NS	Recent diagnosis; changes in treatment
SC-GSC	Opto	OHT; GS; G: stable	Other ocular diseases; G: unstable; recent treatment changes	NS	Unstable G
SC-QMC	Opto	OHT; GS; G: stable	G: severe and complicated; other ocular diseases	NS	Narrow angles

Abbreviations: OHT = ocular hypertension patient; GS = glaucoma suspect patients; G = glaucoma patient; NS = not specified; NS° = not specified, but based on the recommendations of the Australian and New Zealand Glaucoma Interest Group and the Royal Australian and New Zealand College of Ophthalmologists; Opto = Optometrist; OT = Ophthalmic Technician; ONP: Ophthalmic Nurse Practitioner; H = history of; y = year(s); DRP = diabetic retinopathy; mo = month(s); IOP = intra-ocular pressure; PDS = pigment dispersion syndrome; PXF = pseudo exfoliation syndrome; SC = Shared Care clinic; NMS = Non-Medical Staff; ANGIG&RANZCO = recommendations of the Australian and New Zealand Glaucoma Interest Group and the Royal Australian and New Zealand College of Ophthalmologists; SC-RVAC = shared care clinic, established between the Royal Victorian Eye & Ear Hospital and the Australian College of Optometry; CGSC = recommendations from the Canadian Glaucoma Society Committee; GFU = Glaucoma Follow-up Unit; MC = Mayo Clinic’s campus in Rochester; MEH = Moorfield Eye Hospital; QMC = Queen’s medical centre; SGC = Stable Glaucoma Clinic in New Zealand; GMC = Glaucoma Management Clinic in Australia.

**Table 4 jcm-10-04785-t004:** The quality of care provided by the non-medical staff and the quality of care provided by the shared care clinic in general.

SC/Hospital	First Author	Compliance with Protocol (GFU)/Guidelines	Results of Tests and Examinations	Glaucoma Status	Referral	MD: Clinical Management
SC-RVAC	Bentley et al. [41]	SC vs. AAO PPPg, ANGIG&RANZCO - >85%	Optic nerve assessment skills (% correct diagnosis): -mean increase of 14.0% *	NS	NS	NS
SC-GFU	Holtzer-Goor et al. [38]	NMS vs. protocol - >98.8% of the visits	NS	SC vs. STC - % visits stable: SC (17.0%) ≈ StC (16.0%) ** - % visits with shortening of FU-interval: SC (16.0%) ≈ StC (15.1%) **	NMS: correct referral to GE - 84.4% of the remarkable cases	SC vs. StC - Treatment changes: SC (14.0%) ≈ StC (15.0%) **
	Holtzer-Goor et al. [36]	NMS vs. protocol - IOP, VA, GDx: > 97.5% - VF: 25.4% SC/StC vs. protocol - IOP: SC ≈ StC ** - VA: SC > StC * - GDx: SC > StC * - VF: SC ≈ StC **	SC vs. StC - VA decline (% visits): SC (3.9%) < StC (6.3%) * - IOP: SC ≈ StC ** - VF: SC ≈ StC ** - GDx: SC ≈ StC **		NMS: correct referral to seek advice from GE - 100.0%: SOF on GDx/VA- 84.6%: IOP > tIOP - 68.2%: VA declined >2 lines	SC vs. StC -Treatment changes: SC (14.0%) ≈ StC (15.0%) ** -Reason for change: SC ≈ StC **
	Lemij et al. [37]	NMS vs. protocol - IOP, VA, GDx: > 97.5% - VF: 41.2% *** SC/StC vs. protocol - IOP: SC ≈ StC ** - VA: SC > StC * - GDx: SC > StC * - VF: SC ≈ StC ** -Slit-lamp exam: SC < StC *	SC vs. StC - IOP: SC ≈ StC ** - VA: SC ≈ StC ** - GDx: SC ≈ StC ** - VF: SC ≈ StC **		NMS: correct referral to GE (50.0%) - 92.0%: SOF on GDx - 75.0% SOF on VF - 66.7%: IOP > tIOP - 36.0%: VA declined >2 lines	SC vs. StC -Treatment changes: SC (14.1%) ≈ StC (15.4%) ** -Reason for change: SC ≈ StC **
SC-MC	Damento et al. [37]	SC/StC vs. AAO PPPg (mean number of diagnostic tests) - 13 mo: SC > StC * - 25 mo: SC > StC *	NS	SC vs. StC (number of patients visits) - 13 mo: SC > StC * - 25 mo: SC > StC *	NS	NS
	Winkler et al. [35]	SC/StC vs. AAO PPPg (% of patient visits) - Combined compliance *: SC > StC * - VF: SC ≈ StC ** - Gonio: SC > StC * - Fundus photographs: SC > StC * - OCT: SC > StC * - CCT: SC ≈ StC **	NS		NS	NS
	Shah et al. [33]	Opto vs. GE (frequency of clinical test data used to assess progression) - IOP: opto > GE * - Disc hemorrhage: opto≈ GE ** -Fundus photographs: opto≈ GE ** - VF: opto < GE (p=0.07, tendency) - OCT: opto < GE *	Among all HCP (GEs and optos); among GEs only: - IOP: κ = 0.57; κ = 0.57 - Disc hemorrhage: κ = 0.65; κ = 0.59 -Fundus photographs: 77%; 89% - VF: κ = 0.45; κ = 0.47 - OCT: κ = 0.26: κ = 0.51	Among all HCP (GEs and optos); among GEs only: - κ = 0.37; κ = 0.39	NS	NS
SC-MEH	Banes et al. [40]	NS	Opto vs. GE - IOP: OD median difference = -0.25 mmHg, OS median difference = 0.00 mmH - Slit-lamp exam (cup/disc): median difference = 0, greatest difference = 0.15 - VF: κ = 0.80–0.81	Opto vs. GE - FU-interval: κ = 0.97	NS	Opto vs. GE - Medical and surgical treatment: κ = 0.93–1.00
	Banes et al. [42]		Opto vs. GE - Slit lamp exam: sensitivity and specificity ≈ 83% Opto vs. GE; GE vs. GE - VF: κ = 0.37–0.33; κ = 0.39	Opto vs. GE; GE vs. GE - FU-interval: κ = 0.35; κ = 0.41	Opto vs. GE; GE vs. GE - Correct referral to GE: 72.0% agreement; 72.0% agreement	Opto vs. GE; GE vs. GE - “eye drop” treatment: κ = 0.67; κ = 0.74 - cataract surgery: 94.0%; 93.0% - glaucoma surgery: 95.0%; 97.0%
SC-QMC	Ho et al. [39]	NS	Opto vs. GE - VF: κ = 0.81–0.93	Opto vs. GE - next appointment: κ = 0.88–0.97	Opto vs. GE - Correct referral to GE: κ = 0.96–1.00	Opto vs. GE - “eye drop” treatment: κ = 0.96–1.00
SC-SGC	Bhota et al. [43]	NS	NS	NS	Opto vs. GE - Correct referral to GE: 66.1% agreement	NS
SC-GMC	Phu et al. [44]	NS	Opto vs. GE - Gonio: 59.8% agreement on structures (fair to moderate), 93.4% exact agreement with final diagnosis	NS	NS	NS

Abbreviations: SC = shared care clinic; StC = standard care clinic; * = statistical significant difference (*p* ≤ 0.05); ** = no statistical significant difference (*p* > 0.05); κ = kappa; IQR = interquartile range; GE = glaucoma expert; NMS = non-medical staff; Opto = optometrist; HCP = health care providers; IOP = intra-ocular pressure; VA = visual acuity; VF = visual field; Gonio = gonioscopy; OCT = optical coherence tomography; HRT = Heidelberg retinal tomography; GDx = GDx ECC scanning laser polarimetry; CCT = central corneal thickness; Combined compliance* = combined completion of visual field, gonioscopy, measurement of central corneal thickness, and imaging (OCT or fundus photographs); SOF = suspicion of progression; VF: 41.2% ***: Out of the 34 patients who required a visual field examination on a yearly basis, 20 patients did not receive it in the SC-GFU; mo = month(s); AAO PPPg = American Academy of Ophthalmology Preferred Practice Pattern guidelines; ANGIG&RANZCO = recommendations of the Australian and New Zealand Glaucoma Interest Group and the Royal Australian and New Zealand College of Ophthalmologists; SC-RVAC = shared care clinic, established between the Royal Victorian Eye & Ear Hospital and the Australian College of Optometry; GFU = glaucoma follow-up unit; MC = Mayo Clinic’s campus in Rochester; MEH = Moorfield Eye Hospital; QMC = Queen’s medical centre; GHGC = Greenwich hospital glaucoma clinic; SGC = Stable Glaucoma Clinic in New Zealand; GMC = Glaucoma Management Clinic in Australia.

**Table 5 jcm-10-04785-t005:** Baseline Characteristics “Virtual Clinics”.

First Author, Year	Country	Hospital	VC	Study Sample	NG/OSR vs. FU-Patients
Banes et al., 2018	UK	Units from the HES	/	/	NG/OSR and FU-patients
Wright and Diamond, 2014	UK	3 glaucoma clinics: Bristol, Nuneaton and Kingston	GCC	24257 patients	FU-patients
Kotecha et al., 2017	UK	MEH	GSC (GSMS)	1380 patients	NG/OSR
Clarke et al., 2017	UK	MEH	SMS (GSMS)	204 patients	FU-patients
Kotecha et al., 2015	UK	MEH	SMS (GSMS)	1575 patients	FU-patients
Nikita et al., 2019	UK	MEH	SMS (GSMS)	2015 patients	FU-patients
Kotecha et al., 2005	UK	MEH	GSMS	43 patients	NG/OSR and FU-patients
Choong et al., 2003	UK	SH	GAC	100 patients	NG/OSR
Rathod et al., 2008	UK	SH	GAC	78 patients	NG/OSR
Court and Austin, 2015	UK	SH	GAC	170 patients (85 StC and 85 GAC)	NG/OSR
Tatham et al., 2021	UK	PAEP	VC-PAEP	105 patients (55 StC and 50 VC-PAEP)	FU-patients
Gunn et al., 2021	UK	MREH; BEH	VC-MREH; VC-BEH	148 patients	FU-patients
Mostafa et al., 2020	UK	PAEP	VC-PAEP	116 patients	FU-patients
Nikita et al., 2021	UK	MEH	VC-MEH	2017 patients	NG/OSR and FU-patients
Spackman et al., 2020	UK	REIP	VC-REIP	68 patients	FU-patients

Abbreviations: UK = United Kingdom; MEH = Moorfield Eye Hospital; HES = Hospital Eye Services; SH = Singleton Hospital; GCC = Glaucoma Classifying Clinic; GSC = Glaucoma Screening Clinic; SMS = Stable Monitoring Service; GSMS = Glaucoma Screening and Stable Monitoring Service; GAC = Glaucoma Assessment Clinic; VC = virtual clinic; StC = standard care clinic; NG/OSR = new glaucoma/ocular hypertension suspect referrals; FU = follow-up, PAEP = Princess Alexandra Eye Pavilion, MREH = Manchester Royal Eye Hospital, BEH = Bristol Eye Hospital, REIP = The Royal Eye Infirmary Plymouth.

**Table 6 jcm-10-04785-t006:** The prerequisite skills of the non-medical staff working in the corresponding virtual clinic.

VC	First Author	NMS	History Taking	IOP	VA	Slit-Lamp	Von Herick	OCT (Angle)	VF	Fundus Photographs	OCT	HRT	CCT
GCC	Wright and Diamond	Opto OT	x* x*	x* x*	x* x*	x* x*	x* x*	0 0	- x	- x	0 0	0 0	- x
GSC (GSMS)	Kotecha et al.	Clinician; OT	- x*	- x	- x	0 0	0 0	- x	- x	- x	0 0	0 0	- x
SMS (GSMS)	Clarke et al. Kotecha et al. Nikita et al.	ONP OT OT NS	x* - x* NS	x* - x x	- x x x	x(a) - 0 0	NS 0 0 0	0 0 0 0	- x x x	- x x x	0 0 0 x	- x 0 0	0 0 0 0
GSMS	Kotecha et al.	Clinician OT	- NS	- NS	- NS	- NS	- NS	- NS	- NS	- NS	- NS	- NS	- NS
GAC	Choong et al. Rathod et al. Court and Austin	ONP ONP ONP	x x x	x x x	0 0 0	0 x x	0 x NS	0 0 0	x x x	0 x x	0 0 0	0 x x	0 0 0
VC-PAEP	Tatham et al. Mostafa et al.	OT OT ONP	x x x	x x x	NS NS NS	x x x	NS NS NS	0 0 0	x x x	x x x	x x x	x x x	x x x
VC-MREH; VC-BEH	Gunn et al.	OT	x	x	x	0	0	0	x	0	0	x	0
VC-MEH	Nikita et al.	OT	x	x	x	0	0	x*	x	x	0	x	0
VC-RAEP	Spackman et al.	NS	NS	NS	NS	NS	NS	NS	NS	NS	NS	NS	NS

Abbreviations: x = task performed by the corresponding member of the non-medical staff; - = task performed by the other member of the non-medical staff; 0 = task not performed by any member of the non-medical staff; x* = not specified which member of the non-medical staff performed the clinical assessment; Opto = optometrist; OT = ophthalmic technician; ONP: ophthalmic nurse practitioner; HCA = health care assistant; x(a) = anterior segment; NS = not specified; NMS = non-medical staff; IOP = intra-ocular pressure; VA = visual acuity; OCT(angle) = anterior segment optical coherence tomography for angle assessment; VF = visual field; OCT = optical coherence tomography; HRT = Heidelberg retinal tomography; CCT = central corneal thickness; VC = virtual clinic; GCC = Glaucoma Classifying Clinic; GSC = Glaucoma Screening Clinic; SMS = Stable Monitoring Service; GSMS = Glaucoma Screening and Stable Monitoring Service; GAC = Glaucoma Assessment Clinic, PAEP = Princess Alexandra Eye Pavilion, MREH = Manchester Royal Eye Hospital, BEH = Bristol Eye Hospital, REIP = The Royal Eye Infirmary Plymouth.

**Table 7 jcm-10-04785-t007:** The characteristics that render patients suitable or unsuitable for each virtual clinic.

VC	First Author	NG/OSR vs. FU-Patients	Suitable	Unsuitable
GCC	Wright and Diamond	FU-patients	General FU-pool (all types and various stages of risk)	NS
GSC (GSMS)	Kotecha et al.	NG/OSR	First: low risk of developing G Later: up to three risk factors for G	Definitive signs of G; angle closure suspects; IOP > 32 mmHg
SMS (GSMS)	Clarke et al.	FU-patients	OHT; GS; G: stable and low risk; open angle inclusive PDS and PXF	Poor mobility: poor VF; poor disc imaging
	Kotecha et al.	FU-patients	OHT; GS; G: stable and low/moderate risk	Phakic angle closure/suspects; monocular; coexisting ocular comorbidity; best-corrected VA < Snellen 6/12; H glaucoma filtration surgery; concerns regarding adherence; requirement of hospital transport to attend; signs of cognitive impairment
	Nikita et al.	FU-patients	G: most types, various stages of risk	NS
GSMS	Kotecha et al.	NG/OSR;	Low/moderate risk of developing G	NS
		FU-patients	OHT; GS; G: stable and low/moderate risk, open-angle	NS
GAC	Choong et al. Rathod et al. Court and Austin	NG/OSR NG/OSR NG/OSR	All NG/OSR * All NG/OSR * All NG/OSR *	NS NS NS
VC-PAEP	Tatham et al. Mostafa et al.	FU-patients FU-patients	G: mild to moderate stable GS; OHT	H glaucoma filtration surgery; phakic angle closure/suspects
VC-MREH; VC-BEH	Gunn et al.	FU-patients	G; GS; OHT	<18 years of age; unable to speak English
VC-MEH	Nikita et al.	NG/OSR and FU-patients	G (most types); GS; H ocular surgery/glaucoma laser/retinal laser	Unstable advanced G
VC-REIP	Spackman et al.	FU-patients	NS	NS

Abbreviations: FU = follow-up; NG/OSR = new glaucoma/OHT suspect referrals; G = glaucoma patient; OHT = ocular hypertension patient; GS = glaucoma suspect patient; PDS = pigment dispersion syndrome; PXF = pseudoexfoliation syndrome; All NG/OSR * = all risks of developing glaucoma; H = history of; IOP = intra-ocular pressure; VA = visual acuity; VF = visual field; NS = not specified; VC = virtual clinic; GCC = glaucoma classifying clinic; GSC = Glaucoma Screening Clinic; SMS = Stable Monitoring Service; GSMS = Glaucoma Screening and Stable Monitoring Service; GAC = Glaucoma Assessment Clinic; PAEP = Princess Alexandra Eye Pavilion; MREH = Manchester Royal Eye Hospital; BEH = Bristol Eye Hospital; REIP = The Royal Eye Infirmary Plymouth. PCAG = Primary closed-angle glaucoma.

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
