# Peer review of "Shared Care and Virtual Clinics for Glaucoma in a Hospital Setting"

_jcm, 2021, doi:10.3390/jcm10204785_

Round 1

Reviewer 1 Report

An interesting article pointing in the direction of further development.

Reviewer 2 Report

The manuscript by Dr. Simons and coworkers deals with an important topic: how to overcome the problem of very busy clinics and doctors in managing an ever greater number of patients affected by glaucoma? This review suggests that the two different alternative approaches, using non-medical staff, could help overcome this increasingly critical situation. Both the shared care and virtual clinics can give prompt and reliable answers, at least in low-risk glaucoma patients,  in comparison with the current standard care. This type of approach would be very helpful, allowing the busy ophthalmologist to devote more time to more complex cases.

The manuscript is very interesting, well written and exhaustive. The Authors should be congratulated for bringing this important topic, little treated in literature, to the attention of the medical profession, in particular of those who are interested in glaucoma.

I have only few minor points that need to be clarified, which include the following:

  • Line 83: why tele-medicine was excluded? I believe this type of approach could be interesting in this context.
  • Line 206: “OCT” should also be added here.
  • Line 249: a definition of “advanced glaucoma” could be useful.
  • As stated by the Authors at line 438, all the virtual clinics included in this review are located in the UK, therefore the conclusions may not apply in other countries, especially outside the Anglo-Saxon world, where the gap between non-medical staff and doctors is greater. Perhaps this limit could be briefly discussed in the Discussion or in the Conclusion section.

Reviewer 3 Report

The present paper presents an interesting and important. The experimental protocol is clearly described and results are deeply discussed. The conclusions are convincing and well integrated with previous results.
